# QIIME2 enhances multi-amplicon sequencing data analysis: a standardized and validated open-source pipeline for comprehensive 16S rRNA gene profiling

Armando G. Licata,[1] Marica Zoppi,[1] Chiara Dossena,[1] Federico Rossignoli,[1] Davide Rizzo,[1] Manuela Marra,[2] Giorgio Gargari,[3] Giacomo Mantegazza,[3,4] Simone Guglielmetti,[4] Luca Bergamaschi,[5] Olga Nigro,[5] Stefano Chiaravalli,[5] Maura Massimino,[5] Loris De Cecco[1]

**ABSTRACT**    Multi-amplicon sequencing is a cost-effective method for profiling multiple regions of the 16S rRNA gene, offering a more comprehensive view of microbial diversity. However, implementing such pipelines on open-source platforms (e.g., QIIME2) is often hindered by limited documentation and lack of validation against established tools. This lack of standardization poses challenges for researchers, particularly in clinical and experimental settings. This study aims to: (i) develop and benchmark a standardized, open-source QIIME2- and R-based pipeline for 16S rRNA gene profiling using semiconductor-based sequencing, comparing it with a commercial, closed-source software; and (ii) validate its effectiveness in a pediatric cancer cohort to examine parental influence on the microbiome and child-caregiver microbial relationships. We generated 16S rRNA profiles from 5 mock communities and 12 child-caregiver fecal sample pairs. Benchmarking against commercial software showed that the multi-region (V2–9) approach produced microbial profiles nearly identical to proprietary outputs, with higher sequencing depth and improved taxonomic resolution compared to single-region analyses. Both approaches demonstrated similar microbial richness, accurate mock community reconstruction, and high reproducibility ($R = 0.99$, $P < 0.0001$). These findings were further validated using fecal samples. Application of the pipeline to pediatric samples revealed distinct, differentially abundant *Bifidobacterium bifidum* and *Bifidobacterium adolescentis* variants in children whose microbiota closely resembled that of their caregivers. Overall, this study presents a validated, open-source QIIME2 and R pipeline for multi-amplicon sequencing, providing a standardized and reproducible framework for 16S rRNA gene profiling in clinical and research contexts.

**IMPORTANCE** Multi-amplicon sequencing comprehensively characterizes microbial communities by targeting multiple regions of the 16S rRNA gene. However, analytical workflows and reference databases provided by commercial library preparation kits frequently rely on proprietary primers and closed-source pipelines, which can limit transparency, reproducibility, and adaptability. To address these limitations, we developed and validated an open-source bioinformatics pipeline utilizing QIIME2 and R. Our pipeline integrates data from all targeted 16S regions, generating microbial profiles comparable to those produced by proprietary software. Validation was performed using mock samples and fecal samples collected from pediatric cancer patients and their caregivers, confirming the pipeline's reliability and broad applicability. Furthermore, our pipeline enables detailed analysis of microbial variants, surpassing traditional genus-level restrictions and fully leveraging the enhanced coverage offered by multi-amplicon sequencing. Our findings highlight the necessity of adopting open-source solutions to ensure scientific reproducibility and adaptability to emerging methodologies.

Address correspondence to Loris De Cecco, loris.dececco@istitutotumori.mi.it.

The authors declare no conflict of interest.

See the funding table on p. 18.

**KEYWORDS** 16S metagenomics, QIIME2, pediatric tumors, mock community

Recent advancements in amplicon-based microbiome studies enabled researchers to analyze multiple hypervariable regions of the bacterial 16S rRNA gene (16S) or sequence the entire gene directly. This gene serves as a cornerstone in microbial community analysis. It revolutionized the field by circumventing the need for traditional microbiological assays to identify bacterial species, especially those that are unable to culture (1). The 16S rRNA gene, which codes for the 30S subunit of the prokaryotic ribosome, is approximately 1,540 nucleotides long. It is highly conserved across all bacterial species and contains nine hypervariable regions (V1–9), which are the ideal markers for distinguishing between different bacteria. Indeed, the V3 and V4 regions, together, are the gold standard for bacterial identification at the genus level (2). The V1 region effectively distinguishes *Staphylococcus aureus* from coagulase-negative species (3), while the V2 and V3 regions enable identification of all bacterial genera. The V6 region differentiates most species, except those within the Enterobacteriaceae family. In contrast, the V4, V5, V7, and V8 regions are less effective as targets (4).

Amplicon-based sequencing focuses on one or more hypervariable (V) regions within the 16S rRNA gene. In this approach, specific primers amplify the target V region, after which the resulting products are ligated with adapters and indexes, allowing multiple samples to be sequenced simultaneously on various NGS platforms. Although more advanced methods are currently available—such as shotgun sequencing, which characterizes entire genomes, and third-generation sequencing technologies (e.g., Oxford Nanopore and Pacific Biosciences), which can sequence the full 16S gene without biases introduced by multiple amplification steps. Given the relative immaturity of the latest long-read sequencing technologies, at present, amplicon-based sequencing remains a cost-effective and practical choice, particularly for large cohort studies (4, 5). It also bypasses challenges related to host DNA contamination common in low-biomass samples and avoids ethical concerns related to human DNA incidental findings in shotgun data. These features underscore the continued relevance of multi-region 16S approaches in translational microbiome research. However, this technique is limited by primer bias and the inherent challenges in achieving high taxonomic resolution. To address these issues, several multi-region amplicon library preparation kits have been developed, which improve both primer coverage and taxonomic specificity by targeting multiple variable regions of the 16S rRNA gene (6, 7). Examples of such kits include the xGen 16S v2 and ITS1 Amplicon Panel (Integrated DNA Technologies) and the Ion 16S Metagenomics Kit (Thermo Fisher Scientific).

This approach is becoming increasingly significant due to the growing interest in cancer research on "polymorphic microbiomes," a recently suggested hallmark of cancer (8). Microbes can directly cause cancer, influence the host's immune response to foster malignancy, and play a crucial role in shaping the effectiveness of anticancer treatments (8). Microbial contribution is especially pronounced in pediatric contexts, where early-life exposures to microorganisms have a significant influence on long-term health outcomes. Children inherit from their parents not only their genetic makeup but also a shared microbial environment that shapes immune maturation, metabolic health, and disease resilience. These factors hold particular importance for pediatric cancer patients (9). Thus, understanding how parental influences shape a child's microbiome is critical to elucidating its role in immune development and infection susceptibility, thereby informing targeted health interventions (10).

Although the technical workflow behind library preparation has remained the same for years, the analysis of sequencing outputs continues to evolve. Over the years, numerous software tools and pipelines have been proposed, including both closed-source and open-source options. Among the latter, QIIME (11), developed by several members of the human microbiome project consortium, was developed to offer an open-source platform for analyzing 16S data. The subsequent version, QIIME2 (12), shifted from the old operational taxonomic unit (OTU)-based analysis to an amplicon

sequencing variant (ASV)-based approach that resolves sequences to single-nucleotide differences, providing higher resolution and accuracy in microbial community profiling. Despite the original intention of providing a standardized analysis platform, QIIME2 can sometimes present challenges to bioinformaticians. The multitude of possible analysis combinations can make it difficult to select the appropriate pipeline for sequencing data, particularly for multi-amplicon based sequencing data, which lacks comprehensive official documentation. One notable example is the Ion 16S Metagenomics Kit by Thermo Fisher, which allows sequencing of six hypervariable regions in a single library using their proprietary primer sets. This technology provides an end-to-end solution with an integrated analytical workflow and has been proposed to be routinely applied in clinical settings for diagnostic purposes (13–15). In addition, an increasing wealth of evidence has proved the value of Ion 16S Metagenomics in the oncology field, highlighting the microbiota in cancer pathogenesis and response to therapy (16–18).

To date, the official reference standard for analyzing sequencing data obtained from the Ion16S Metagenomics Kit (Thermo Fisher Scientific) is Ion Reporter (IR), a closed-source platform specifically designed to analyze Ion Torrent data and mixed orientation sequences from various proprietary library preparation kits, including the 16S Metagenomics Library Prep Kit. The outputs from this pipeline are limited and poorly documented. For instance, the platform does not support phylogeny computation—crucial for analyses such as UniFrac (19) or Faith's phylogenetic diversity (PD) (20)—resulting in a loss of valuable information affecting the accuracy of evolutionary relationships and lineage diversification among microbial taxa. The only published paper discussing strategies for analyzing this data using QIIME2 is by Maki et al. (21). Although this study provides valuable insights into the application of QIIME2 with Ion16S data, it lacks a direct benchmark against the official IR workflow, leaving its comparative performance unassessed.

Explainability of the analytical steps involving data handling and analysis is crucial for the reliable deployment of 16S findings into clinical actionable assay. The primary goal of this study is to develop and validate a standardized, open-source bioinformatics pipeline for Ion Torrent 16S rRNA gene profiling. We benchmarked a modified version of Maki et al.'s workflow (21) against the IR platform to ensure accurate reconstruction of mock communities and robust characterization of microbial compositions across multiple hypervariable regions. Additionally, we integrated a complementary analysis framework in R, providing a versatile toolkit for downstream investigations. To further validate this pipeline and explore the influence of parental microbiota on pediatric gut communities, we applied it to a cohort of pediatric cancer patients and their caregivers, ultimately aiming to inform microbiome-targeted therapeutic strategies in clinical settings.

## MATERIALS AND METHODS

### Libraries preparation

For this study, we generated independent 16S rRNA amplicon libraries from: (i) mock samples using the ZymoBIOMICS Microbial Community DNA Standard (Zymo Research); (ii) stool samples from a clinical study as validation. The DNA from the ZymoBIOMICS standard was diluted to 5 ng/µL. For amplification of the 16S hypervariable regions, 2 µL of diluted DNA was used as input in reactions with the 16S Metagenomics Kit (Thermo Fisher Scientific), following the manufacturer's protocol, which included 18 PCR cycles. For stool samples, DNA was first extracted using the QIAsymphony DSP/Virus/Pathogen Midi Kit (Qiagen) and then adjusted to an input of 100 ng before proceeding with the same 16S hypervariable region amplification protocol. The libraries were then prepared using the Ion Plus Fragment Library Preparation Kit (Thermo Fisher Scientific). Library quantification was performed using the dsDNA HS Assay Kit on Qubit fluorometer (Thermo Fisher Scientific), and quality was checked using the 4150 TapeStation automated electrophoresis system (Agilent Technologies). Finally, the libraries from each

replicate were pooled at 50 pM. The sequencing was performed on an Ion 530 barcoded chip with an Ion GeneStudio S5 Prime sequencer system (Thermo Fisher Scientific).

## Metataxonomic analysis using IR and QIIME2 on mock samples

### IR workflow

To establish a reference taxonomic profile for our mock and fecal samples, we utilized the IR software to generate an OTU table, which was subsequently compared to the QIIME2 pipeline (Fig. 1). Specifically, the sequencing data were demultiplexed, converted to unmapped BAM (uBAM) format, and uploaded into Ion Reporter v.5.20 (Thermo Fisher Scientific). On this platform, each mock uBAM file was analyzed using the unmodified Metagenomics 16S w1.1 workflow. This integrated, closed-source pipeline performs OTU picking and employs a curated version of the Greengenes database (v.13_5) for taxonomic assignment.

### Sequences deconvolution and data import on QIIME2

The sequencing data were first demultiplexed, and each FASTQ file—containing reads from multiple hypervariable regions—was deconvoluted into separate files for each V region using the MetagenomicsPP Deconvolution Tool (Thermo Fisher Scientific), as previously described (21, 22). This tool, available as a plugin for the sequencer, separates reads based on the amplified region by recognizing proprietary primer sequences. The resulting V region-specific FASTQ files were then imported into QIIME2 (version 2023.7), which was run in a Docker container to ensure reproducibility.

### Denoising and ASV picking

Denoising and ASV inference were performed using the DADA2 algorithm, following the pipeline proposed by Maki et al. (21). To account for the specific error profile of Ion Torrent sequencing, the "-pyro" option was enabled, as recommended in the DADA2 documentation (https://benjjneb.github.io/dada2/faq.html). Default parameters were used unless otherwise specified. Reads were truncated at the position corresponding to a Phred33 quality score ≥25 at the 25th percentile of the quality distribution, yielding final read lengths between 170 and 210 base pairs. This quality filtering approach ensured a balance between read retention and accuracy in downstream ASV calling.

### Phylogenetic tree reconstruction

A phylogenetic tree was constructed using the fragment insertion SATé-enabled phylogenetic placement (SEPP) algorithm (23), available as a QIIME plugin, with the curated phylogeny reference from Greengenes (v.13_5). SEPP was used specifically to address the challenges of short sequencing fragments associated with Ion Torrent data. Unrelated sequences that were not recognized or used to reconstruct the phylogenetic tree were discarded. The feature table was then filtered to remove these ASVs.

### Taxonomy classification

The Greengenes database (v.13_5) was selected due to compatibility with the IR system, enabling direct benchmarking. Despite its limitations in species-level classification, it remains a practical standard for genus-level analyses in clinical settings. This database can be accessed at the Greengenes database website. In the QIIME2 environment, representative ASV sequences for each region were taxonomically annotated using a region-specific custom-trained database. This database was based on reference sequences extracted from Greengenes using universal primers specific to the regions (V1–V2, V3–V4, V4, V6–V8, V7–V9, and V2–9), as detailed in Table 1. The database files (fasta files and taxonomy) were imported into QIIME2, where a naïve Bayes classifier was trained for each region, resulting in five classifiers trained with region-specific primers. The filtered ASV sequences for each V region were then taxonomically annotated using the

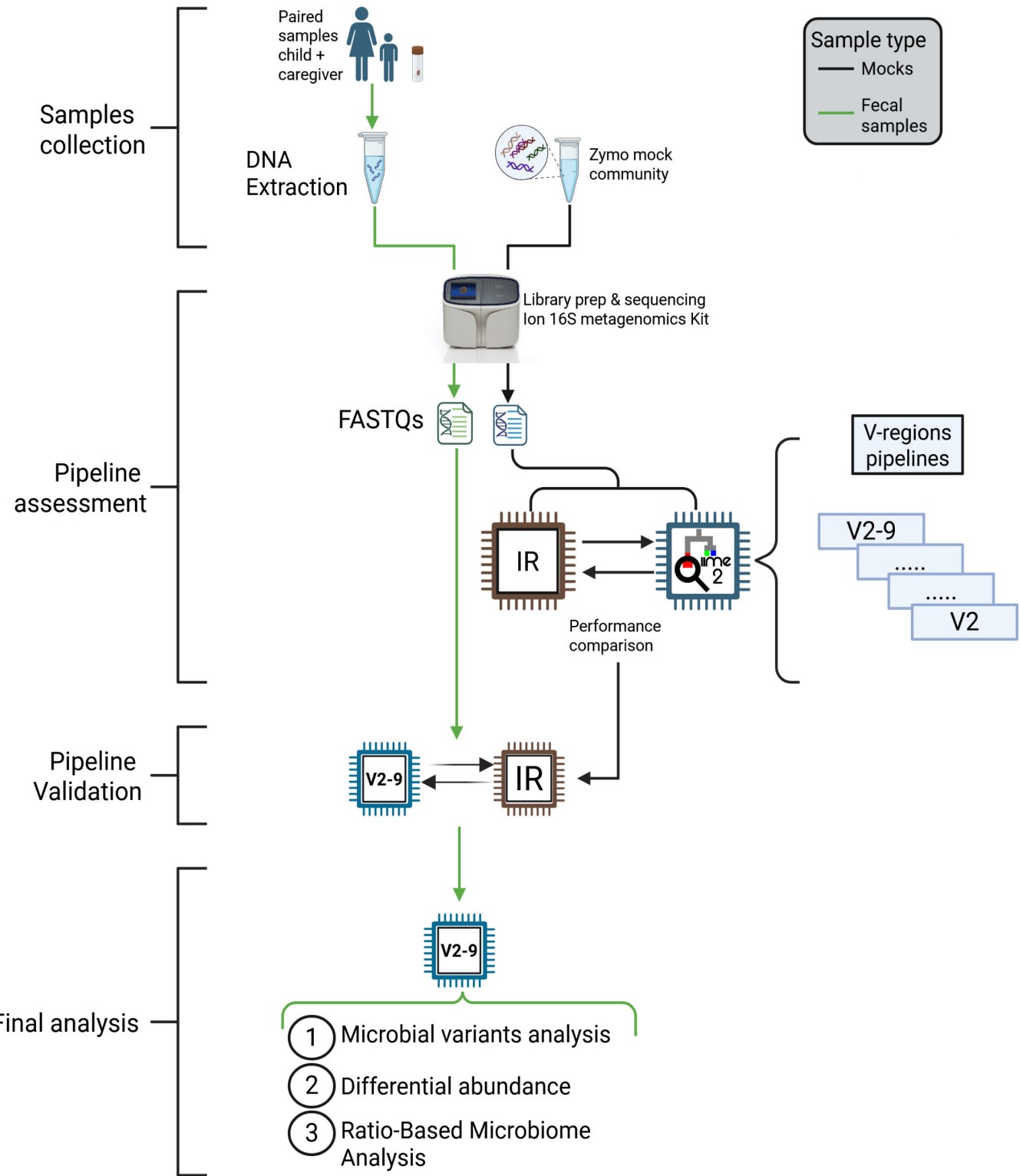

**FIG 1** Overall study design and bioinformatic analysis workflow. Overview of the experimental and analytical framework used in this study. Paired stool samples from children and their caregivers underwent DNA extraction, while the Zymo mock community, provided as ready-to-use DNA, served as a reference standard. All samples were prepared for sequencing using an Ion 16S metagenomics kit. Resulting FASTQ files were processed through a comparative pipeline assessment step, in which multiple V-region pipelines (V2–9) were evaluated to ensure robust performance. Following validation, the selected workflows were applied to the full data set, culminating in final analyses that encompassed microbial variant profiling, differential abundance testing, and ratio-based microbiome assessments.

**TABLE 1** V-region-specific forward and reverse primers and sequence length parameters used for the extraction of region-specific reference sequences from Greengenes and Silva 16S databases

| V-region | Forward primer | Reverse primer | Forward sequence (5′–3′) | Reverse sequence (5′–3′) | Specificity | Sequences length (min – max [nt]) | Reference |
|---|---|---|---|---|---|---|---|
| V1-V2 | 27F | 338R | AGA GTT TGA TYM TGG CTC AG | GCT GCC TCC CGT AGG AGT | Universal | 300–320 | (24) |
| V3-V4 | 341F | 785R | CCT ACG GGN GGC WGC AG | GAC TAC HVG GGT ATC TAA TCC | Universal | 430–460 | (25) |
| V4 | 515F | 806R | GTG CCA GCM GCC GCG GTA A | GGA CTA CHV GGG TWT CTA AT | Universal | 280–300 | (26) |
| V6-V8 | 939F | 1378R | GAA TTG ACG GGG GCC CGC ACA AG | CGG TGT GTA CAA GGC CCG GGA ACG | Bacterial | 430–450 | (27) |
| V7-V9 | 1115F | 1492R | CAA CGA GCG CAA CCC T | TAC GGY TAC CTT GTT ACG ACT T | Bacterial | 360–390 | (28) |
| V2–V9 | Bact-27F | Univ-1492R | AGR GTT TGA TCM TGG CTC AG | RGY TAC CTT GTT ACG ACT T | Bacterial/universal | 1,400–1,460 | (29, 30) |

previously trained classifier with the "*classify-sklearn*" algorithm (https://scikit-learn.org/stable/modules/naive_bayes.html#multinomial-naive-bayes). Finally, the filtered feature table, the rooted tree,regarding the taxonomy annotations were obtained and used for downstream analyses.

## Multi-region analysis

Since the IR pipeline generates data from all sequenced regions, a similar approach was used using QIIME2. Specifically, denoised sequences and count tables from all regions were merged to create a comprehensive data set that covers all sequenced regions from the kit. The merging was performed using the "*qiime feature-table merge*" algorithm with the "*--p-overlap-method*" sum option enabled. Sequence files were merged using the "*qiime feature-table merge-seqs*" command. A phylogenetic tree was reconstructed following the previously described method, while taxonomy classification was performed using the VSEARCH algorithm (31). VSEARCH is particularly well-suited for handling high-diversity regions, efficiently processing large data sets, and supporting chimera detection and removal, which enhances the accuracy of taxonomic classification. The reference sequences used for classification were generated using the V2–9 primers as previously described (Table 1).

To generate the Silva-based results, we downloaded the SILVA v.138-99 reference database from the QIIME2 resources page (https://docs.qiime2.org/2023.7/data-resources/). Using the same V2–9 primers (Table 1), we extracted target regions from the SILVA reference by running "*qiime feature-classifier extract-reads*," followed by taxonomic assignment with "*qiime feature-classifier classify-consensus-vsearch*" under default parameters. This SILVA-based taxonomy was then applied to both mock and fecal samples.

## Data import in R software

The comparison between IR and QIIME outputs and the theoretical mock composition was conducted using R software (v.4.2.2). IR's results were downloaded, and the OTU table at genus and family levels was imported as a *.tsv* file using the Tidyverse package (v2.0.0), along with the theoretical microbial composition available at the ZymoResearch website (https://zymoresearch.eu/products/zymobiomics-microbial-community-dna-standard). For the QIIME outputs—features table, taxonomy, metadata, and the rooted tree—these were imported into R using the qiime2R package (v0.99.6) via the "*qza_to_phyloseq*" function, which interprets QIIME's *.qza* files and creates a Phyloseq object. Phyloseq (v1.42.0) (32), a package commonly used to analyze and manipulate microbiome data in R, was employed to create a Phyloseq object for each region. These objects were then merged, and to mitigate biases that can arise when comparing the ASV-based data from QIIME with the OTU-based data from IR. The ASVs from the merged Phyloseq object were collapsed at the genus and family levels using the "*tax_glom*" function built into Phyloseq. Finally, the data set at the genus level was generated by merging the QIIME, IR, and theoretical compositions to proceed with the analyses. Alpha and beta diversity metrics were computed using Phyloseq. Genus and family level data sets were normalized using the total sum scale method to compare QIIME and IR data with the theoretical composition available in percentage values. (Table S1) provides a summary of key R functions and packages used to import and analyze QIIME2 data in R.

## Mock sample

To establish a standardized 16S rRNA sequencing analysis pipeline, five replicates of mock microbial DNA standards were analyzed using QIIME2, benchmarked against IR and the theoretical Zymo mock composition (Table 2). Deconvoluted regions in QIIME2 (V2, V3, V4, V6–V7, V8, and V9) were independently processed, and a combined data set (V2–9) was generated by merging intermediate feature tables and sequence files from all regions.

**TABLE 2** Theoretical composition and reference metrics for each species in the ZymoBIOMICS Microbial Community DNA Standard (Zymo Research)

| Species | Theoretical composition (%) | | | | |
|---|---|---|---|---|---|
| | Genomic DNA | 16S Only | 16S and 18S | Genome copy | Cell number |
| *Pseudomonas aeruginosa* | 12 | 4.2 | 3.6 | 6.1 | 6.1 |
| *Escherichia coli* | 12 | 10.1 | 8.9 | 8.5 | 8.5 |
| *Salmonella enterica* | 12 | 10.4 | 9.1 | 8.7 | 8.8 |
| *Lactobacillus fermentum* | 12 | 18.4 | 16.1 | 21.6 | 21.9 |
| *Enterococcus faecalis* | 12 | 9.9 | 8.7 | 14.6 | 14.6 |
| *Staphylococcus aureus* | 12 | 15.5 | 13.6 | 15.2 | 15.3 |
| *Listeria monocytogenes* | 12 | 14.1 | 12.4 | 13.9 | 13.9 |
| *Bacillus subtilis* | 12 | 17.4 | 15.3 | 10.3 | 10.3 |
| *Saccharomyces cerevisiae* | 2 | NA | 9.3 | 0.57 | 0.29 |
| *Cryptococcus neoformans* | 2 | NA | 3.3 | 0.37 | 0.18 |

## Validation data set of fecal samples: study description and ethics statement

Given the taxonomic complexity of fecal microbiota compared to mock communities and its associated bioinformatic challenges, we benchmarked the performance of our proposed pipeline (IR vs QIIME2) using an external validation data set. This data set comprised 16S rRNA gene sequencing profiles from stool samples collected in a clinical study from June 2020 to July 2022. The original study, named INT 77/20 (22), characterized gut microbiota dynamics in pediatric cancer patients and their cohabiting caregivers, defined as adults providing parental care (hereafter "caregivers"), including biological parents or non-parental guardians. For validation purposes, we analyzed 12 patient-caregiver pairs from this cohort using the bioinformatic and statistical pipeline described here, repurposing these data to assess methodological generalizability independent of the original study's clinical aims.

The original cohort included pediatric cancer patients aged 3–22 years (mean ± SD = 14 ± 5 years) with diagnoses spanning lymphoma, central nervous system tumors, neuroblastoma, and bone/soft tissue sarcomas. All participants were antibiotics- and steroids-naïve at enrollment. Detailed clinical metadata were recorded, including weight, height, BMI, symptoms at enrollment, gender, age, primary disease, concomitant diseases, tumor treatment received, and time since enrollment. Ethical approval was granted by the local committee (INT77/20), and written informed consent was obtained from participants or their caregivers, as appropriate for age.

## Statistical analysis

Statistical analyses were performed to evaluate differences in microbial communities. The Kruskal-Wallis test was applied to evaluate the significance of different alpha diversity metrics and the relative abundance of various genera between regions on a univariate basis. For pairwise comparisons of the relative abundance of genera between different regions, Dunn's post hoc test from the rstatix R package (v.0.7.2) was utilized. To infer significant compositional differences in microbial communities between V regions and analysis methods, and to compare fecal samples analyzed with IR and QIIME2, a permutational multivariate analysis of variance (PERMANOVA) with 9,999 permutations was conducted using the *adonis2* function from the vegan package (v.2.6-4) on Bray-Curtis distance matrices. Pearson's correlation test, implemented via the *cor.test* function built into R-base (v.4.2.2), was employed to assess the correlation of microbial profiles between IR and QIIME2 for both the mock and fecal data sets. For categorizing pediatric patients into high- and low-similarity groups, pairwise Pearson's correlations of microbial abundances were computed for each child-caregiver pair, and Euclidean distances were calculated in the principal coordinate space derived from UniFrac-based PCoA. Pediatric samples exhibiting both a significant correlation (*P* value ≤ 0.05) and a Euclidean

distance below the median value (0.255) across all samples were classified into the high-similarity group; the remaining samples were designated as low-similarity group. ASVs significantly differing between these groups were identified through differential abundance analysis using the negative binomial distribution method implemented in DESeq2 (33) (v.1.38.3), followed by Wald's test with false discovery rate (FDR)-adjusted *P* values (FDR ≤ 0.05 was considered significant). The representative sequence for each differentially enriched ASV was aligned with BLASTn against the NCBI microbial 16S rRNA gene database. Taxa that showed a 100% match in species taxonomy were manually annotated based on the results. To assess the similarity of distance matrices generated from different taxonomic reference databases, we performed Procrustes analyses on corresponding ordinations (34). Specifically, we calculated Bray-Curtis distance matrices for each data set, conducted principal coordinate analysis (PCoA), and then aligned the resulting configurations using the "*Procrustes*" function in the *vegan* R package. The degree of fit was quantified by the Procrustes sum of squares ($m^2$), and statistical significance was evaluated via the associated permutation test "*protest.*" Shorter connecting segments between samples in the aligned plots denote a closer match between the two ordinations, thereby facilitating both a clear visual and quantitative assessment of the influence of the reference database on overall community structure.

## RESULTS

### Mock community analysis

To assess the performance of different 16S rRNA gene-based analytical pipelines in accurately representing microbial community structure, we initially compared sequencing depth and relative genus abundances across all targeted regions (Fig. 2A). Rarefaction curves indicate that both the IR and V2–9 data sets afforded superior coverage compared to individual regions in capturing microbial diversity, achieving total sequencing depths of approximately 20,000–30,000 reads. Conversely, individual regions exhibited lower sequencing depths (~10,000 reads) and did not attain saturation, thereby limiting feature detection. Although the IR, V2, and V2–9 profiles detected seven of the eight expected genera in the Zymo mock community, the genus *Escherichia* was initially undetected but subsequently confirmed in the V2–9 data set at an average relative abundance of 8.67% (SD ± 0.44%, FDR = n.s.) following the application of an updated an updated Silva v.138-99 database (Fig. S1). Notably, all profiles also identified additional genera that were absent from the reference standard (Fig. 2B). Specifically, *Cronobacter* was identified only in the IR data set, exhibiting a significantly lower relative abundance (0.06%, SD ± 0.09%, FDR = 0.008) compared to all other genera (Fig. 1C; Table S2). Similarly, *Klebsiella* was detected only in the merged V2–9 data set, with an abundance of 0.31% (SD ± 0.08%) that was significantly lower than that of the actual present genera (FDR < 0.01) (Fig. 2C; Table S2). Despite the comprehensive coverage provided by the IR and V2–9 data sets, these false positives persisted, albeit at minimal abundance.

Overall, although the IR and V2–9 data sets generally recapitulated the anticipated mock composition, individual regions—particularly V8 and V9—exhibited significant deviations. These findings underscore not only the inherent limitations of single-region analyses in capturing comprehensive microbial diversity but also reveal that the standard IR pipeline, partially due to its reliance on the outdated Greengenes v.13_5 database, may fail to detect certain genera. Consequently, this emphasizes the critical importance of utilizing updated databases, such as Silva v.138-99, for enhanced accuracy in microbial profiling.

### Evaluation of microbial diversity and reproducibility in IR and QIIME2 pipelines

We compared the outputs of IR and QIIME2 pipelines using additional analytical methods to evaluate compositional changes and assess result reproducibility (Fig. 3). Alpha

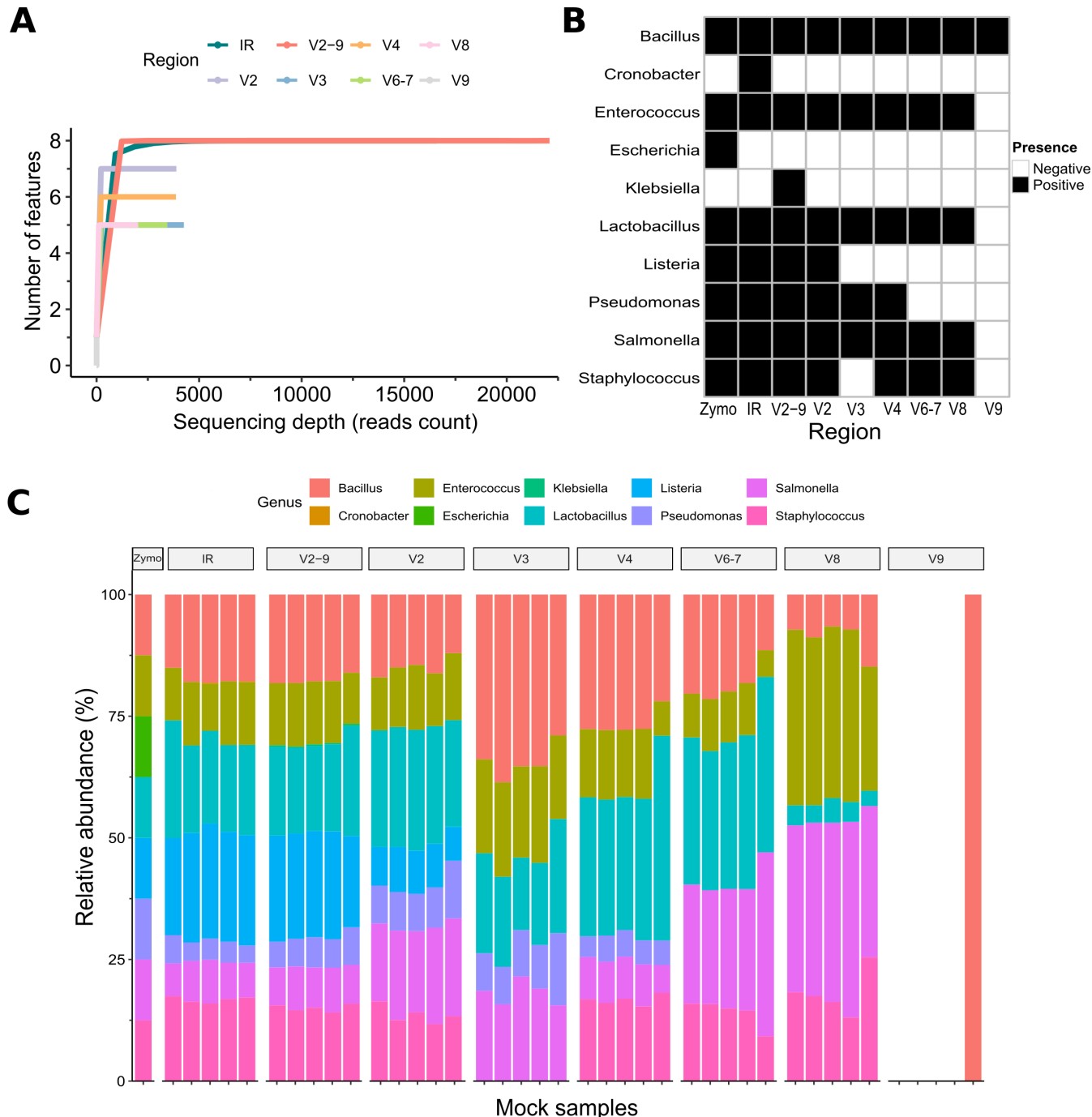

**FIG 2** Comparison of microbial composition and sequencing coverage across different regions and pipelines in mock samples. (A) Rarefaction curves showing the number of observed features for various sequencing regions (IR, V2, V2–9, V3, V4, V6–V7, V8, and V9) across different sequencing depths. Notably, the IR and V2–9 data sets achieve greater coverage and exhibit higher alpha diversity than the individual regions. (B) Heatmap illustrating the presence (black) or absence (white) of genera across the sequencing regions. Rows correspond to specific genera, while columns represent different regions. (C) Stacked bar plot displaying the relative abundance of bacterial genera in mock samples, including the Zymo mock community, across the targeted regions. The color-coded profiles reveal variability in microbial composition, with the IR and V2–9 data sets most closely matching the Zymo mock community.

diversity analysis was calculated using Observed, Shannon, and Simpson indices, revealing significant differences in microbial richness and evenness across single-region data sets (Fig. 3A). Although false-positive genera were detected in the IR and V2–9 data sets, both exhibited significantly higher alpha diversity compared to all other

single-region data sets. Notably, no significant differences in richness and evenness were observed between IR and V2–9, with both closely approximating the composition of the Zymo mock community (*P* value > 0.05, Table S3). To further assess microbial community similarity, Bray-Curtis distances were computed and visualized using PCoA (Fig. 3B). The PCoA revealed significant compositional differences among the single-region data sets, whereas the IR and V2–9 samples exhibited substantial overlap, as corroborated by

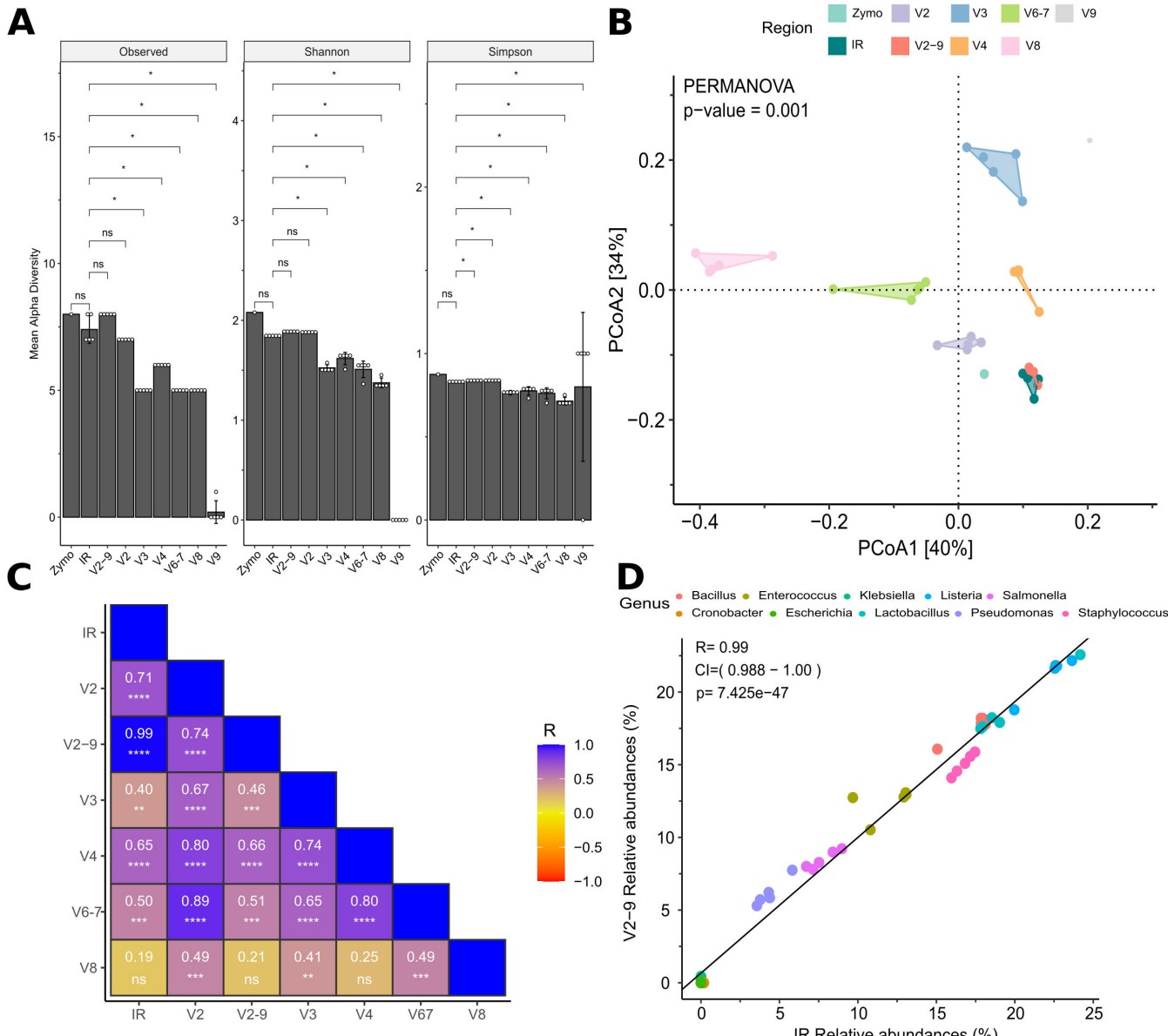

**FIG 3** Comparison of alpha diversity, beta diversity, and correlation of microbial profiles across sequencing regions. (A) Alpha diversity metrics (Observed richness, Shannon, and Simpson indices) for each tested sequencing region (IR, V2, V2–9, V3, V4, V6–7, V8, and V9) and the Zymo mock community. Statistical significance was assessed using Dunn's post hoc test; significant differences are indicated by asterisks (ns = not significant; *P* value < 0.05; **P* value < 0.01). Notably, the IR and V2–9 data sets exhibit significantly higher alpha diversity compared to single-region data sets. (B) PCoA based on Bray-Curtis distances, with polygons illustrating beta dispersion for each region. PERMANOVA analysis (*P* value < 0.001) reveals distinct microbial compositions among single-region data sets, except for V2–9 and IR, which do not differ significantly (PERMANOVA *P* value > 0.05). ANOVA indicates no significant differences in beta dispersion across regions. (C) Heatmap of Pearson correlation coefficients (R) displaying the similarity in microbial composition among different sequencing regions. Color intensity corresponds to the magnitude of Pearson's R values, with warmer colors representing lower correlation. (D) Scatter plot of genus-level relative abundances in V2–9 vs IR data sets, demonstrating a near-perfect correlation (Pearson's *R* = 0.99, *P* value < 0.0001). This strong linear relationship underscores the concordance in microbial profiles between these two sequencing strategies.

PERMANOVA (Table S4). Additionally, the beta dispersion analysis demonstrated reproducibility among replicates (Table S5). To identify the QIIME2 pipeline that closely aligns with the bacterial composition derived from the IR pipeline, we correlated the relative abundance of each genus across all data sets (Fig. 3C). Overall, the single-region data sets were significantly correlated with the IR-derived microbial abundance (Pearson's $R > 0.4$, $P$ value $< 0.01$), with the notable exception of the V8 data set, which exhibited no significant correlation. Reads mapping to the V9 region were undetected in four out of five replicates, resulting in near-zero counts across almost all samples. Consequently, the V9 data set was excluded from further analysis. Interestingly, the V2–9 data set exhibited the highest correlation with IR microbial composition (Pearson's $R = 0.99$, $P$ value $< 0.0001$, Fig. 3D), suggesting that the V2–9 region most accurately reflects microbial community structure.

## Validation of the QIIME pipeline using fecal samples from a cohort of pediatric cancer patients and their caregivers

Encouraged by the mock-community results, we extended our comparison between the IR pipeline and QIIME 2 analyses across all 113 stool samples from the INT-77/20 cohort (22) (Fig. 4). As observed with the mock community analysis, sequencing-depth distributions were highly consistent between IR and the multi-region V2–9 pipeline, while all single-region pipelines yielded significantly lower read counts (Fig. S2). IR and V2–9 analyses remained virtually indistinguishable: richness and evenness indices (Observed, Shannon, and inverse-Simpson) showed no significant differences (pairwise Wilcoxon tests, $P > 0.05$; Fig. 4A), and their Bray-Curtis PCoA coordinates nearly completely overlapped (Fig. 4B). Single-amplicon V2, V4, and V6–V7 pipelines demonstrated similar behavior, whereas V3, V8, and V9 introduced systematic shifts that distinctly separated them from the IR/V2–9 cluster. Sample-wise Pearson correlations reinforced this pattern (Fig. 4C and D). IR-derived profiles closely matched V2–9, V2, V4, and V6–V7 pipelines, consistently showing near-identical taxon-abundance patterns (median $-\log 10$ $P$ adj. $> 6$ and median $R > 0.90$). In contrast, V8 exhibited only moderate similarity, and V3 and V9 displayed minimal to no concordance, with several samples even yielding negative correlations. Additional analyses comparing taxonomic assignments using Silva (v.138-99) and Greengenes (v.13_5) databases, at both genus and ASV levels, further confirmed robust concordance (Fig. S3). Collectively, these findings validate the reliability and reproducibility of the QIIME 2 V2–9 pipeline and underscore the consistent performance of both methods in accurately characterizing microbial community structures.

## Influence of caregiver-child microbiome similarity on gut microbiota composition in pediatric cancer patients

After validating our V2–9 pipeline against IR microbial profiles, we utilized QIIME2's unique analytical capabilities on a controlled subset from the INT 77/20 cohort. Specifically, QIIME2 uniquely enables analyses incorporating both phylogenetic distances and amplicon sequence variant (ASV)-level resolution, features that are not available with the IR workflow. The selection criteria for this controlled subset included matched pairs of pediatric patients and their parental caregivers who were naïve to antibiotic and steroid-based medications, resulting in a paired cohort comprising 12 patient-caregiver pairs. This subset was analyzed to characterize the microbiota comprehensively at both phylogenetic and ASV levels (Fig. 5).

Initially, PCoA was conducted on microbial abundance data using UniFrac distances, which account for evolutionary relationships among taxa to capture variations in community structure (Fig. 5A). Based on the UniFrac-based PCoA and abundance correlations, the 12 pediatric samples were stratified into high-similarity ($n = 6$) and low-similarity ($n = 6$) groups reflecting the degree of phylogenetic and abundance similarity to their respective caregivers' microbiota (Fig. 5B).

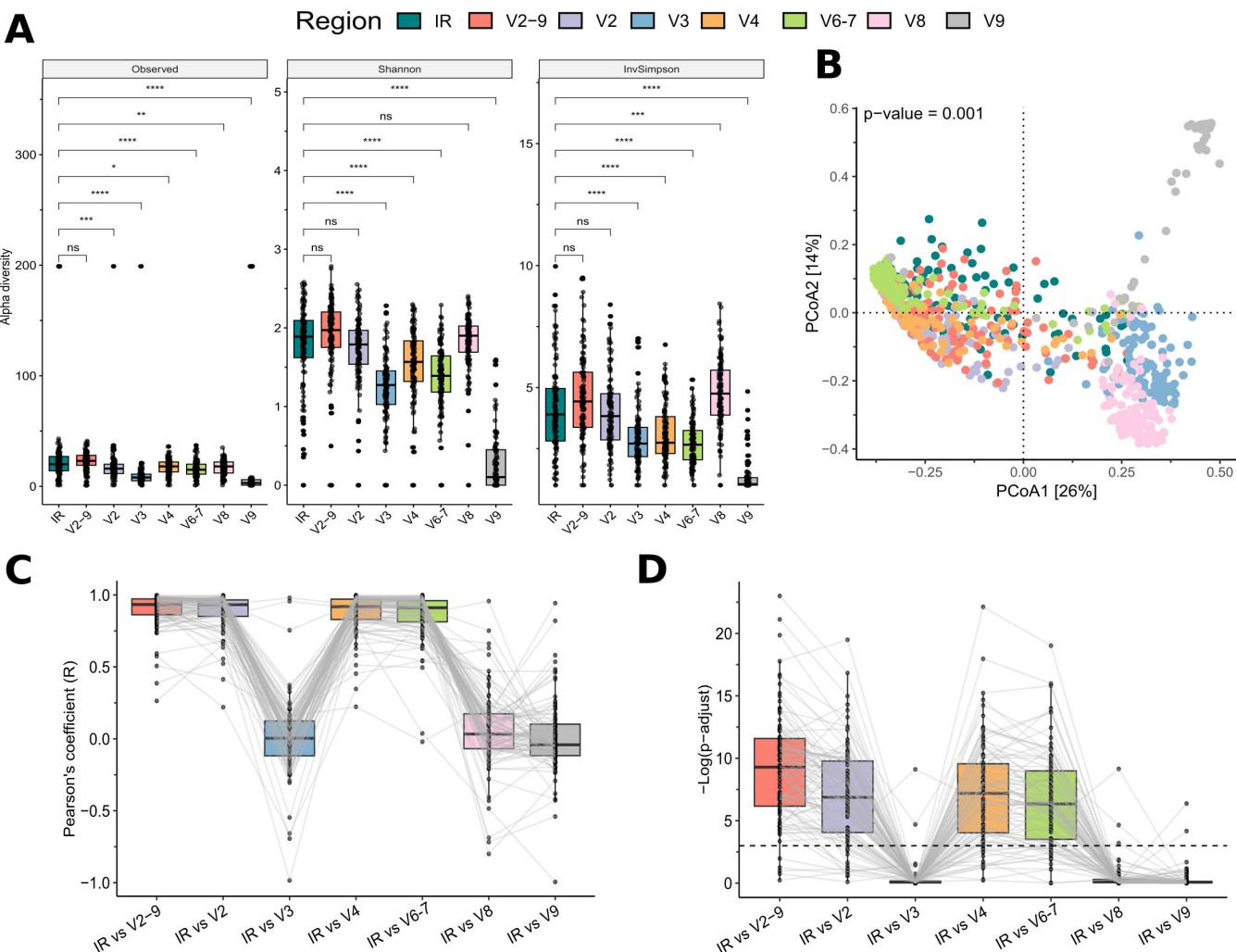

**FIG 4** Comparative evaluation of microbial community profiling by Ion Reporter (IR) and QIIME 2 pipelines across multiple amplicon regions. Comparison of microbial profiles generated by IR and various QIIME 2 single- and multi-region pipelines. (A) Alpha diversity indices (Observed, Shannon, and inverse-Simpson) showing richness and evenness across the different pipelines, with statistical significance determined by pairwise Wilcoxon tests. (B) Bray-Curtis PCoA plot depicting the clustering of samples based on pipeline-derived microbial profiles. (C) Pearson correlation coefficients comparing IR vs QIIME 2 pipelines across samples. (D) Statistical significance (−log10 adjusted *P* values) of Pearson correlations, highlighting concordance differences between pipelines. (C and D) Paired samples are connected by gray lines.

Subsequently, alpha diversity within the pediatric samples was assessed using two key metrics: Faith's PD, which incorporates phylogenetic relationships among taxa, and the number of observed features. The analysis revealed that the high-similarity group exhibited significantly higher Faith's PD and a greater number of observed features compared to the low-similarity group (Wilcoxon test, *P* < 0.05; Fig. 5C). This finding suggests that pediatric patients whose microbiota more closely resemble that of their caregivers tend to have a richer and more diverse microbial community.

To further elucidate the differences between these groups, we performed a differential abundance analysis using DESeq2. This analysis identified 20 ASVs that were significantly differentially abundant between the high- and low-similarity groups (|Log2 fold change| > 1, FDR < 0.05; Fig. 5D). Approximately two-thirds of these ASVs were enriched in the high-similarity group, whereas the remaining ASVs were more abundant in the low-similarity group. Notably, two ASVs exhibiting 100% sequence identity to *Bifidobacterium bifidum* and *Bifidobacterium adolescentis* were enriched in the high-similarity group, indicative of both higher abundance and higher prevalence within this

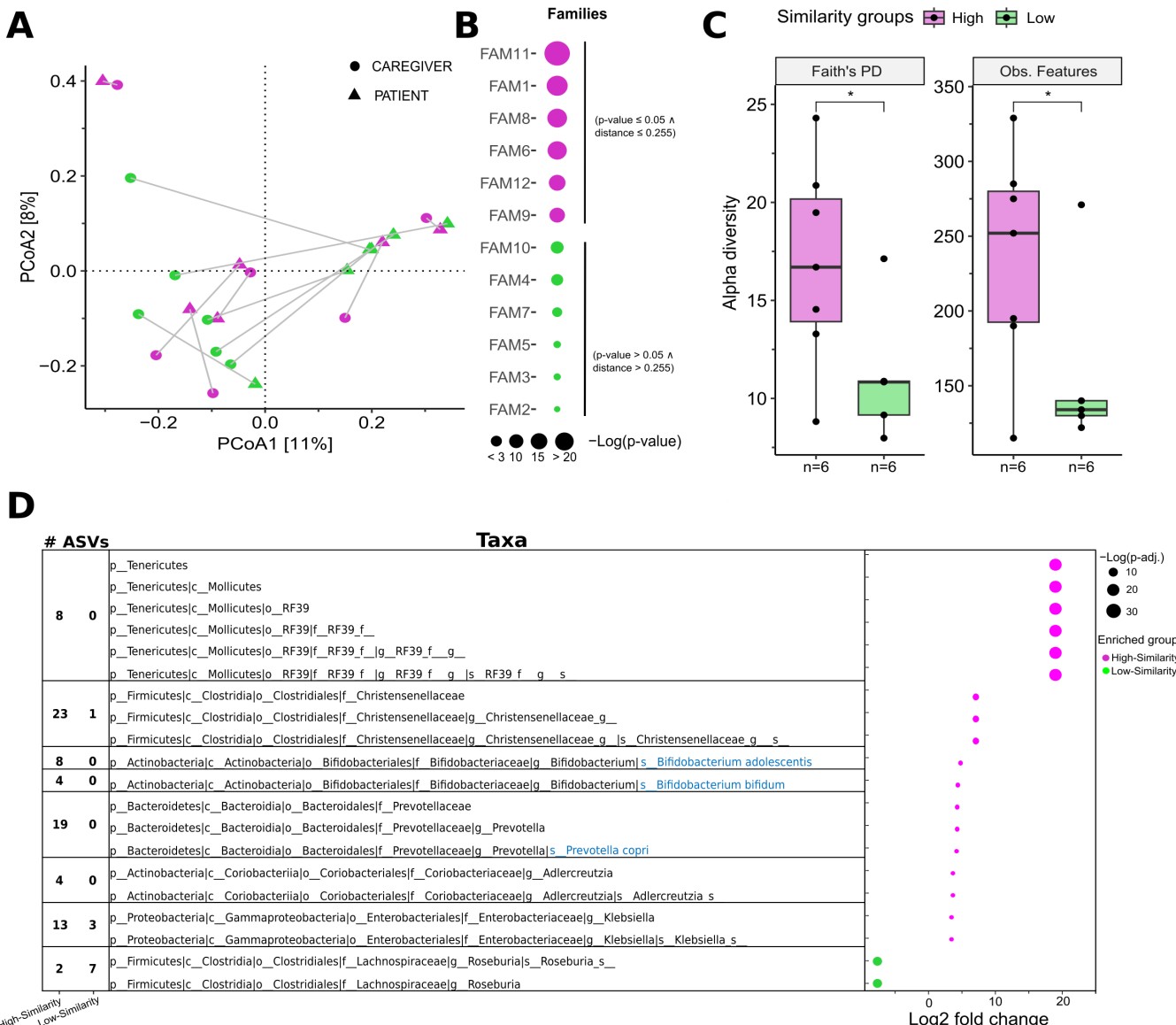

FIG 5 Influence of caregiver microbiota on pediatric patient gut microbiome composition. (A) PCoA based on UniFrac distances, illustrating the gut microbial community structure of pediatric patients (triangles) and their caregivers (circles). Each child-caregiver pair is connected by a gray line. Pediatric samples were designated as high‐ (violet) or low‐similarity (green) depending on whether their correlation with the caregiver was significant (*P* value ≤ 0.05) and their Euclidean distance in the PCoA space fell below 0.255. (B) Bubble plot showing the correlation strength (Pearson's correlation) between child and caregiver microbiomes for each family. Bubble size indicates the −log(*P* value), reflecting statistical significance, while bubble color corresponds to whether each family falls into the high‐ (violet) or low‐similarity (green) group. (C) Box plots depicting two alpha diversity metrics (Faith's phylogenetic diversity and observed features, i.e., ASV richness) for pediatric samples in the high‐ and low‐similarity groups. (D) Differential abundance analysis using DESeq2, comparing high‐ (violet) versus low‐similarity (green) groups. Significantly differentially abundant ASVs (|Log2FoldChange| > 1, FDR < 0.05) are listed by taxonomic affiliation alongside the number of ASVs per taxon. Blue‐highlighted species names indicate >99% sequence identity to entries in the NCBI 16S rRNA database, confirming species‐level identification. In the bubble plot, circle size represents −log10(FDR), and color indicates the similarity group in which the ASV is enriched.

cohort, whereas several ASVs within the genus *Roseburia* were more prevalent in the low-similarity group.

Collectively, our analyses revealed that the groups were enriched with specific genus variants, suggesting an association between the degree of microbiota similarity with the caregiver and the microbial composition observed in pediatric patients. Moreover, these findings underscore that alterations in ASV prevalence—reflecting the frequency of

detection—may be as critical as changes in abundance for distinguishing the microbial profiles of high- versus low-similarity groups.

## DISCUSSION

An essential prerequisite for translating assays to clinical practice is the establishment of robust and reproducible analytical workflows that enhance explainability, interpretability, and ultimately, acceptability of the assay's findings. In recent years, microbiota research has attracted considerable attention across various biomedical fields, including oncology. However, computational pipelines must be tailored to the specific field of application. This is particularly critical in oncology, where fine-tuning analytical methods is a critical step to ensure that identified taxonomic profiles accurately represent the underlying microbial communities. For example, Thermo Fisher's Ion 16S Metagenomics solution offers an end-to-end solution with an integrated analytical workflow within the IR platform. Although this platform is user-friendly and easy to implement in clinical settings, its acceptability is hindered by its nature as "black box" software which constrains transparency in data processing.

In this study, we conducted a comparative analysis of a QIIME2 and Phyloseq-based pipeline versus Thermo Fisher's 16S Metagenomics workflow available in the IR platform using two distinct data sets and methodological approaches: (i) a data set comprising five independent mock communities, and (ii) a cohort of 113 matched fecal samples derived from pediatric cancer patients and their caregivers. These samples were processed using two newly proposed and thoroughly documented pipelines designed to assist bioinformaticians in evaluating different V-region profiles (21). However, despite their comprehensive documentation, the lack of benchmarking against the expected results from IR, the official software for analyzing Ion 16S metagenomics data, raised concerns regarding their practical applicability (35). To address this, we conducted a detailed evaluation of these pipelines to determine the most effective strategy for handling this data exclusively with open-source tools like QIIME2 and Phyloseq in R.

It is well recognized that mock communities often fall short in capturing the complexity and diversity of gastrointestinal microbiomes. To overcome this limitation, we validated our approach using 113 fecal samples, which more accurately reflect real-world biological variability. To establish a robust benchmark, we processed each bio-sample using both QIIME2 and IR. However, this was limited by IR's requirement for uBAM files directly from the sequencing platform, which restricted the size of our cohort. Overall, a few single-regions data sets analyzed with QIIME2 accurately recapitulated the expected mock genus composition, a trend also observed with other methods analyzing Ion Torrent data (36). Specifically, the V2 detected seven out of eight genera present in the mock community without introducing false positives, albeit with a lower sequencing depth. In contrast, both IR and the V2–9 data set identified the same number of mock genera with a substantially higher sequencing depth, providing more reads for analysis. Nevertheless, these approaches also detected false positives, such as *Klebsiella* and *Cronobacter*—two genera not present in the mock community but observed at limited abundances (0.01–0.3%). Interestingly, the genus *Escherichia* was undetected using the standard IR pipeline, which relies on the outdated Greengenes 13.5 database. To investigate this issue, taxonomic classifications were reassigned with the updated Silva 138-99 database (covering the V2–9 regions), while maintaining the same alignment algorithm. This reanalysis successfully identified the *Escherichia–Shigella* genus, thereby resolving the initial discrepancy. Nevertheless, the species *Escherichia coli*, known to be present in the mock community, remained undetected across all data sets, with the family Enterobacteriaceae consistently identified instead. Notably, these observations are in line with findings from other studies employing similar library preparation kits and pipelines dependent on Greengenes v.13.5 (36).

Detailed analyses were performed to identify the favored patterns capable of reproducing the profiles generated by IR. Our results indicate that both alpha diversity (i.e., species richness within samples) and beta diversity (i.e., differences between

samples) metrics identify the V2–9 as the sole configuration that aligns with the expected outcomes. To further evaluate the performance of QIIME2, we compared the presence and relative abundance of each genus across all V-region profiles with those obtained from IR. Remarkably, only the V2–9 data set achieved a correlation coefficient ($R$) of 0.99, demonstrating an almost perfect correlation. These findings substantiate that integrating multiple regions of the 16S gene significantly enhances the accuracy of microbial profiling.

Subsequently, the V2–9 pipeline was assessed on a data set of 113 fecal samples from the INT-77/20 cohort, a real-world study in pediatric oncology (22) to validate the overlap initially observed with mock communities. Sequencing depth was highly consistent between V2–9 and IR, while all single-region pipelines yielded significantly lower read counts. Overall, V2–9 produced microbial profiles closely matching those of IR, with indistinguishable richness and evenness metrics and nearly identical Bray-Curtis ordination patterns, as well as significantly similar and similarly abundant feature lists. In contrast, pipelines based on V3, V8, and V9 introduced systematic shifts and lower concordance with IR. These variations in single-region performance are not unexpected, as specific 16S regions differ in their ability to resolve particular taxa. Consequently, agreement with IR or V2–9 depends on the extent to which region-specific amplicons capture dominant genera within the studied microbial environment (37, 38). Additional comparisons using Silva (v.138-99) and Greengenes (v.13_5) databases confirmed broadly concordant community structures, although minor discrepancies were noted. These are likely attributable to differences in taxonomic resolution and the outdated nature of Greengenes (39), which remains in use for IR via a manually curated version, whereas our analyses used the original publicly available release. Collectively, these findings reinforce the robustness and reproducibility of the V2–9 pipeline for accurately profiling diverse microbiomes, indicating that any observed discrepancies primarily reflect differences in database curation rather than the pipelines themselves (39). A second edition of Greengenes was released in 2022 (40). However, when the identical Greengenes (v.13_5) reference is applied to both pipelines, the overall profiles remain consistent.

Finally, the V2–9 pipeline was applied to investigate the gut microbiome in a subset of 12 pediatric cancer patient-caregiver pairs, all with no history of antibiotic or steroid treatment. The analysis revealed that caregiver-child relationships significantly influence microbial similarities within family units, accounting for a substantial portion of the variation among samples ($R^2$ = 0.55, PERMANOVA $P$ < 0.001). This finding is consistent with previous studies suggesting that close physical contact and shared environments contribute to microbiome similarities among family members (34). Although some families displayed stronger microbiome similarities than others, potentially due to variations in diet, lifestyle, genetics, or environmental exposures, the overall trend underscored the significant role of shared environments (4). It is important to acknowledge that the small cohort size constrained our ability to perform broader comparisons across additional clinical variables. Nonetheless, this cohort provided a valuable validation set for the application and evaluation of QIIME2‐based pipelines, while also enabling a preliminary exploration of shared gut microbiome features between pediatric oncology patients and their caregivers, useful for setting up further clinical studies.

Furthermore, patients in the "high-similarity" group were enriched with bacterial ASVs belonging to *B. bifidum* and *B. adolescentis*, whereas the "low-similarity" group exhibited higher abundances of *Roseburia*. Notably, bifidobacteria, particularly *B. bifidum*, are highly host-adapted microorganisms with limited viability outside their host and have been documented to be vertically transmitted from mother to child (41–43). Consequently, they have been shown to exhibit significant co-diversification with their respective host (44). In contrast, members of the genus *Roseburia*, such as *R. hominis*, generally display minimal evidence of co-phylogeny with their host (44), suggesting that in the "low-similarity group," direct bacterial transmission between patient and caregiver may be substantially reduced.

Collectively, these results underscore that the integration of multiple 16S regions with open‑source workflows provides an effective strategy for characterizing the gut microbiome in pediatric oncology settings, while also highlighting the potential influence of caregiver-child interactions on microbial community composition.

## Conclusion

In summary, our study demonstrates that the V2–9 QIIME2 pipeline reliably replicates the performance of Thermo Fisher's proprietary software IR for multi-amplicon 16S rRNA gene sequencing. Through benchmarking against mock communities and validation with complex fecal microbiomes from pediatric cancer patients and their caregivers, we established that the V2–9 approach provides superior accuracy in microbial profiling. The pronounced microbial similarities within caregiver-child pairs emphasize the influence of shared environments on gut microbiota composition, highlighting the need to account for the family microbial ecological environments in microbiome studies. These findings support the adoption of open-source, standardized pipelines for comprehensive microbiome analysis and lay the groundwork for future investigations into microbiome transmission mechanisms. However, larger cohort studies are needed to further elucidate these mechanisms and advance microbiome-targeted therapeutic strategies for pediatric oncology.

### ACKNOWLEDGMENTS

We acknowledge the patients and their families, as well as the staff at the Pediatric Oncology Unit (i.e., data managers and nurses) who supported the research. We would like to thank Silvana Canevari for her critical revision of the manuscript.

The present study was supported by Bain Capital Children's Fund Europe Covid-19 Appeal (project Cross-Domain interactions among SARS-CoV-2, microbiome and host in pediatric cancer patients) and the Italian Ministry of Health/Regione Lombardia (project ID NET-2019-12371188; All-ages malignant glioma-Holistic management in the personalized minimally invasive medicine era: from lab to rehab—GLI-HOPE).

Data analysis: A.G.L. Data collection: A.G.L., M.Z., C.D., and D.R. Experimental implementation: A.G.L., M.Z., C.D., F.R., D.R., G.G., G.M., S.G., and M. Marra. Interpretation of the data: A.G.L. and L.D.C. Provision of the study materials: L.B., S.C., O.N., and MauM. Study conception: A.G.L. and L.D.C. Study design: A.G.L., L.D.C., and M. Massimino. Study supervision: L.D.C. Visualization: A.G.L. Writing—original draft: A.G.L. annd L.D.C.; Revising manuscript: all authors.

The authors confirm that they did not receive any funding or support from Life Technologies, Inc., a Thermo Fisher Scientific brand, and the company did not participate in the preparation of this manuscript. Furthermore, this manuscript does not promote or endorse any of the company's products.

### AUTHOR AFFILIATIONS

[1]Integrated Biology of Rare Tumors, Department of Experimental Oncology, Fondazione IRCCS Istituto Nazionale dei Tumori, Milan, Italy

[2]NGS Unit-Core Facilities Technical-Scientific Service, Istituto Superiore di Sanità, Rome, Italy

[3]Department of Food, Environmental and Nutritional Sciences, University of Milan, Milan, Italy

[4]µbEat Lab, Department of Biotechnology and Biosciences (BtBs), Università degli Studi di Milano-Bicocca, Milan, Italy

[5]Pediatric Oncology Unit, Fondazione IRCCS Istituto Nazionale dei Tumori, Milan, Italy

### AUTHOR ORCIDs

Armando G. Licata  http://orcid.org/0000-0001-5785-5163

Simone Guglielmetti ⓘ http://orcid.org/0000-0002-8673-8190
Loris De Cecco ⓘ http://orcid.org/0000-0002-7066-473X

## FUNDING

| Funder | Grant(s) | Author(s) |
| --- | --- | --- |
| Bain Capital Children's Fund Europe | | Maura Massimino |
| Italian Ministry of Health/Regione Lombardia | ID NET-2019-12371188 | Maura Massimino |

## AUTHOR CONTRIBUTIONS

Armando G. Licata, Conceptualization, Data curation, Formal analysis, Investigation, Methodology, Visualization, Writing – original draft, Writing – review and editing | Chiara Dossena, Formal analysis, Writing – review and editing | Federico Rossignoli, Formal analysis, Writing – review and editing | Davide Rizzo, Formal analysis, Writing – review and editing | Manuela Marra, Formal analysis, Writing – review and editing | Giorgio Gargari, Formal analysis, Writing – review and editing | Giacomo Mantegazza, Data curation, Writing – review and editing | Simone Guglielmetti, Formal analysis, Writing – review and editing | Luca Bergamaschi, Resources, Writing – review and editing | Stefano Chiaravalli, Resources, Writing – review and editing | Maura Massimino, Funding acquisition, Resources, Writing – review and editing | Loris De Cecco, Conceptualization, Funding acquisition, Investigation, Methodology, Supervision, Writing – original draft, Writing – review and editing.

## DATA AVAILABILITY

The pipeline for analyzing Ion Torrent data using QIIME2, supporting both single-region and multi-region analyses, is available at https://github.com/ArmandoLicata/QIIME2-MultiRegion-Microbiome-Pipeline. This repository also provides a fully dockerized version of QIIME2 v.2023.7. All processed data and details of the bioinformatic steps (including parameter settings, scripts, and environment configurations) are included within the repository documentation. Raw 16S rRNA sequencing data, deconvoluted by region, have been deposited in the NCBI Gene Expression Omnibus (GEO) and are publicly available under accession number GSE300047.

## ADDITIONAL FILES

The following material is available online.

### Supplemental Material

**Supplemental figures and tables (Spectrum01673-25-s0001.docx).** Fig. S1 to S3 and Tables S1 to S6.

### Open Peer Review

**PEER REVIEW HISTORY (review-history.pdf).** An accounting of the reviewer comments and feedback.

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
