## [Reviewer comments · Microbiology Spectrum]

Microbiology Spectrum

QIIME2 enhances multi-amplicon sequencing data analysis: a standardized and validated open-source pipeline for comprehensive 16S rRNA gene profiling

Armando Licata, Marica Zoppi, Chiara Dossena, Federico Rossignoli, Davide Rizzo, Manuela Marra, Giorgio Gargari, Giacomo Mantegazza, Simone Guglielmetti, Luca Bergamaschi, Olga Nigro, Stefano Chiaravalli, Maura Massimino, and Loris De Cecco

Corresponding Author(s): Loris De Cecco, Fondazione IRCCS Istituto Nazionale dei Tumori

Review Timeline:

Submission Date:	June 5, 2025
Editorial Decision:	June 13, 2025
Revision Received:	June 23, 2025
Accepted:	June 24, 2025

Editor: Huanyu Wang

Reviewer(s): The reviewers have opted to remain anonymous.

Transaction Report:

DOI: <https://doi.org/10.1128/spectrum.01673-25>

Re: Spectrum01673-25 (QIIME2 enhances multi-amplicon sequencing data analysis: a standardized and validated open-source pipeline for comprehensive 16S rRNA gene profiling)

Dear Dr. Loris De Cecco:

Thank you for the privilege of reviewing your work. Below you will find my comments, instructions from the Spectrum editorial office, and the reviewer comments.

I am pleased to inform you that your manuscript has been editorially accepted for publication. However, there are a few additional questions in the submission form that need to be answered before the final decision. Once these are completed, please return your submission so that I can move your paper forward to acceptance.

Revision Guidelines

Sincerely,
Huanyu Wang
Editor
Microbiology Spectrum

We thank the Editor for the comments. We have modified the manuscript according to their suggestions, and we hope that they find this new version improved and suitable for publication.

Please find here below a point-by-point response.

1) Upload point-by-point responses to the issues raised by the reviewers in a file named "Response to Reviewers":

The point-by-point responses to reviewers have been uploaded in a separate file titled "Response to Reviewers", not included in the cover letter.

2) Upload a compare copy of the manuscript (without figures) as a "Marked-Up Manuscript" file:

A "Marked-Up Manuscript" file has been uploaded. Figures have been removed as requested, and **figure legends have been placed at the end** of the file.

3) Upload a clean .DOC/.DOCX version of the revised manuscript and remove the previous version:

A clean DOCX version of the revised manuscript (without figures, with figure legends at the end) has been uploaded. The previous version has been removed.

4) Each figure must be uploaded as a separate, editable, high-resolution file (TIFF or EPS preferred), and any multipanel figures must be assembled into one file:

All figures have been uploaded as **separate, high-resolution TIFF files**. Multipanel figures have been **assembled into single files** as required.

5) Any supplemental material intended for posting by ASM should be uploaded with their legends separate from the main manuscript. You can combine all supplemental material into one file (preferred) or split it into a maximum of 10 files with all associated legends included:

Supplemental material has been prepared and uploaded **as a single file**, with legends included and kept **separate from the main manuscript**, as per instructions.

6) Additional updates:

A statement regarding the **submission of the data to GEO** has been added to the manuscript.

Re: Spectrum01673-25R1 (QIIME2 enhances multi-amplicon sequencing data analysis: a standardized and validated open-source pipeline for comprehensive 16S rRNA gene profiling)

Dear Dr. Loris De Cecco:

Your manuscript has been accepted, and I am forwarding it to the ASM production staff for publication. Your paper will first be checked to make sure all elements meet the technical requirements. ASM staff will contact you if anything needs to be revised before copyediting and production can begin. Otherwise, you will be notified when your proofs are ready to be viewed.

Sincerely,
Huanyu Wang
Editor
Microbiology Spectrum